# Revisit Recurrent Attention Model
# from an Active Sampling Perspective

**Jialin Lu**
Simon Fraser University

## 1 Introduction

We revisit the Recurrent Attention Model (RAM, Mnih et al. (2014)), a recurrent neural network for visual attention, from an active information sampling perspective.

The RAM, instead of processing the input image for classification in full, only takes a *glimpse* at a small patch of the image at a time. The recurrent attention mechanism learns where to look at to obtain new information based on the internal state of the network. After a pre-defined number of glimpses, RAM finally makes a prediction as output. Compared with the attention mechanism which now dominates the AI/NLP research such as Transformer (Vaswani et al., 2017) and BERT (Devlin et al., 2018), this recurrent attention mechanism is fundamentally different, as it is used to obtain new information (active sampling of information), rather than processing information that is already fully observed.

In this paper, we identify three weaknesses of this widely-cited approach. First, the convergence of RAM training is slow. Second, RAM does not support dynamic number of glimpses per sample, but uses a fixed number of glimpses for every sample. Third and perhaps most importantly, the performance of the original RAM does not improve but rather decrease dramatically if it takes more glimpses, which is weird and against intuition.

We provide a simple solution of adding two extra terms in the objective function of RAM, insipred from neuroscience research (Gottlieb, 2018) which discusses the logic and neural substrates of information sampling policies in the context of visual attention and gaze. Base on the evidence available so far, Gottlieb (2018) suggested three kinds of motives for the active sampling strategies of decision-making while the original RAM only implements one of them. We incorporate the other two motives in the objective function, and by doing so we 1) achieve much faster convergence and 2) instantly enbale decision making with a dynamic number of glimpse for different samples with no loss of accuracy. 3) More importantly, we find that the modified RAM generalizes much better to longer sequence of glimpses which is not trained for.

## 2 Related work

**Recurrent Attention Model (Mnih et al., 2014).** RAM combines a recurrent neural network with reinforcement learning to utilize visual attention for classification and is trained with policy gradient in a end-to-end manner. At each timestep, it takes a glimpse on a patch of the input image, processes the information and then uses its internal state to select the next location to focus. By adaptively selecting a sequence of patches of images and only processing the selected patches, it outperforms a convolutional neural network baseline while its computation cost is cheap and independent of the input image size. Due to page limit, we refer readers to the original paper (Mnih et al., 2014), online tutorials and high-starred implementation on Github.

The recurrent attention mechanism allows RAM to actively obtain new information for decision making. Note that though RAM is trained end-to-end through the help of the REINFORCE rule (Williams, 1992), it is not a standard reinforcement learning problem where the objective is to maximize the cumulative reward (or, minimize the cumulative regret); instead, RAM maximizes the simple reward (minimizes the simple regret). The reward is given only to the final prediction action,

33rd Conference on Neural Information Processing Systems (NeurIPS 2019), Vancouver, Canada.

but not to the glimpse actions. In a sense, a glimpse action is rewarded indirectly only if it obtains useful information that helps making a correct prediction. This is very similar to the problem of pure-exploration bandits (Bubeck et al., 2009), specifically best arm identification. We think this distinguishing feature of active sampling of information for decision making is not well addressed in deep learning, in reinforcement learning, as well as in neuroscience research (Gottlieb and Oudeyer, 2018; Gottlieb, 2018).

**Weaknesses of Recurrent Attention Model (Mnih et al., 2014).** First, the original RAM paper did not mention the exact convergence of RAM training. We inspect some Github implementations of RAM, one implementation (conan7882, 2018) takes 1000 epochs to achieve the paper accuracy; another (kevinzakka, 2017) claims to reach the paper accuracy in 30 epochs, but we find that it uses a technique to augment the test time accuracy, introduced in the followed work (Ba et al., 2014) of RAM, by processing an input multiple times and averaging the prediction. When we disable this technique in (kevinzakka, 2017), we find that the convergence is in fact much slower.

Second, RAM uses a fixed number of glimpses for each input sample, however we would like to see RAM terminates automatically as long as it is very certain about the prediction. It is possible that some samples are easier to predict and only need fewer glimpses. In the future work section of Mnih et al. (2014), the authors briefly mentioned that they trained a separate network to predict when to terminate. By this way RAM learns to do dynamic prediction "*once it has enough information to make a confident classification*".

Third, RAM does not generalize well when using a larger number of glimpses. This generalization issue comes in two different settings.

- When the number of glimpses is the same for training and testing, which is the experiment setting in the original RAM paper. The authors showed that RAM performs the best in MNIST trained with $N = 6$ glimpses per sample and more glimpses begin to weaken the accuracy.

- When the number of glimpses is not the same for training and testing. We set the number of glimpses to $N = 6$ to train the RAM and test the performance on $N \neq 6$. We find that the accuracy does not improve but decrease dramatically (see Figure 2).

In this paper we evaluate the second setting.

**Implications from (Gottlieb, 2018).** In (Gottlieb, 2018), the authors reviewed recent studies on attentional learning, and highlighted the role of active sampling through visual attention and gaze for decision making. Particularly, Gottlieb (2018) suggested three types of motives for implementing active sampling policies, "*One motive relates simply to the extent to which information is expected to increase the operant rewards of a task; a second motivation is related to reducing the uncertainty of belief states; and yet a third motive may be related to the intrinsic utility or dis-utility of anticipating a positive or a negative outcome (savoring or dread).*" In the context of RAM for classification, only the first motive is implemented by RAM as the reward of correct classification.

## 3 Modification of RAM

We propose to add two extra terms which represents the second and third motives mentioned above, on the original objective function of RAM.

We denote the MNIST dataset of in total $M$ sample-label pairs as $\{X, Y\} = \{x, y\}^M$; the number of class in $Y$ is $K = 10$. the parameter of the RAM architecture is denoted as $\theta$, the output $K$-dimension softmax vector of RAM as $f_\theta(x)$, and the prediction of classification as $\hat{y} = argmax_i f_\theta(x)_i$.

**The original objective function.** Under the distribution of all possible sequences $s_{1:N}$ of actions (at length $N$), RAM maximizes the reward of correct classification as the objective function: $J_{original}(\theta) = E_{p(s_{1:N};\theta)}[\mathbb{1}_{\hat{y}=y}]$ where $p(s_{1:N}; \theta)$ depends on the policy.

Mnih et al. (2014) employed two tricks to maximize $J_{original}(\theta)$ which is non-trival. First, Mnih et al. (2014) borrowed the REINFORCE rule (Williams, 1992) to maximize $J_{original}(\theta)$ through policy gradient. Second, because the policy gradient may have a high variance, Mnih et al. (2014) trained an auxiliary network (one fully-connected layer) called baseline network taking the internal

state as input to mimic the value function, which helps the training. So the true objective function the orignal RAM maximizes is $J_{original} + J_{auxiliary}$.

**New objective function.** We consider the new objective function to be

$$J_{new} = J_{original} + J_{uncertainty} + J_{intrinsic} + J_{auxiliary}$$

We use $J_{uncertainty}$ to model the motive to reduce uncertainty. However, since the uncertainty is not available, we use a similar network like the baseline network to mimic the residual of error. In other words, we use another auxiliary network to predict the different between the classification and the ground truth, taking the internal state of RAM and the current prediction $f_\theta(x)$ as input. We call this predicted diffrence as self-uncertainty, denoted as $u(x)$, which is also a $K$-dimensional vector. Since we wish to minimize this uncertainty, we have $J_{uncertainty} = -\lambda_1 \sum_i |u(x)_i|$

We use $J_{intrinsic}$ to model the intrinsic utility of anticipating outcome. The intrinsic utility of antici- pating outcome is here interpreted as the intrinsic preference over certain prediction outcomes. By intrinsic, it means this preference is inherently built-in regardless of the hidden state or the gathered information and should make RAM prefer to predict a certain class if there is no enough informa- tion. As MNIST is class-balanced and no class is more important than the others, we choose to enforce this preference equally to all the classes, i.e. all the possible prediction outcomes. Techni- cally, the intrinsic preference should encourage the output softmax vector $f_\theta(x)$ to be close to a prior distribution, which is simply uniform. We incorporate this intrinsic belief as the cross entropy to a uniform distribution on the output $K$-dimension vector $f_\theta(x)$: $J_{intrinsic} = -\lambda_2 \sum_i -\frac{1}{K} \log f_\theta(x)_i$. This should regularize RAM to not make any over-confident prediction if there is no enough in- formation. This should be helpful, especially in the early stage of training, when the current active information sampling policy is not so good so that no useful information is gathered.

Note that for simplicity, we merge the objective of the self-uncertainty network and the objective of the baseline network into $J_{auxiliary}$. The entire modified RAM could be trained end-to-end just as the orignal one.

**How to do dynamic decision-making in test-time.** The introduction of self-uncertainty has a side-effect. It gives us an opportunity to enable a dynamic number of glimpses for diffrence samples because now we have an uncertainty measure. Borrowing ideas from pure-exploration bandits, we use self-uncertainty to construct a upper and lower 'bound' for each class, and let RAM terminate if the lower bound of the highest class is higher than the upper bound of the rest classes. Given a timestep $t$, denote the prediction as $f_\theta^t(x)$ and the self-uncertainty as $u^t(x)$, the upper and lower bound for a class $i$ is $Upper(i) = f_\theta^t(x)_i + \beta * u^t(x)_i, Lower(i) = f_\theta^t(x)_i - \beta * u^t(x)_i$ where the ex- ploration rate $\beta$ is a hyperparameter. We take $i^*$ to be the class of the highest $i^* = argmax_i f_\theta^t(x)_i$. RAM terminates when the following condition is met: $Lower(i^*) > \max_{i \neq i^*} Upper(i)$. Given a larger $\beta$, RAM will take more glimpses and when $\beta = 0$, RAM will terminate in only one glimpses.

## 4 Evaluation and Conclusion

We evaluate on MNIST dataset as in the orignal RAM paper. We set the train-time number of glimpses $N = 6$ for it achieves the best test-time accuracy in Mnih et al. (2014). Implementation details see the source code [1].

We first show in Figure 1 that the two new terms in the objective both contribute to a faster conver- gence. We test four cases 1) the orignal objective, 2) add the $J_{intrinsic}$, 3) add $J_{uncertainty}$, 4) add both new terms. We see in Figure 1 that both of our new objective in isolation help a faster learning and together give the fastest convergence.

---

[1]Code: `https://github.com/LuxxxLucy/recurrent-attention-model-revisited`

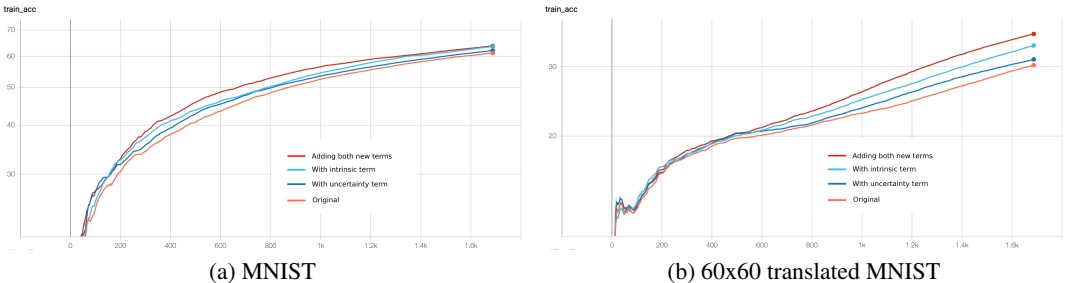

(a) MNIST                                   (b) 60x60 translated MNIST

Figure 1: Train Accuracy during traing (first epoch)

As in Figure 2, we test the trained models with varying number of glimpses. (We want to emphasize that the focus is not the absolute performance , but rather the generalization on more glimpses than train time.) We fisrt evaluate the non-dynamic case (fixed number for all samples). The performance of the original RAM decrease dramatically when $N > 10$. Adding both terms, the modified RAM does not suffer the decrease anymore even when $N$ is large . Also, it is interesting that adding only the uncertainty term, we observe the improvement is very slight and the intrinsic term effectively stablizes the prediction accuracy given more glimpses. We also test the dynamic case by varying the exploration rate. We see that dynamic number of glimpses does not hurt the performance very much, which confirms with the hypothesis that some samples are easier to discriminate and thus need fewer glimpses.

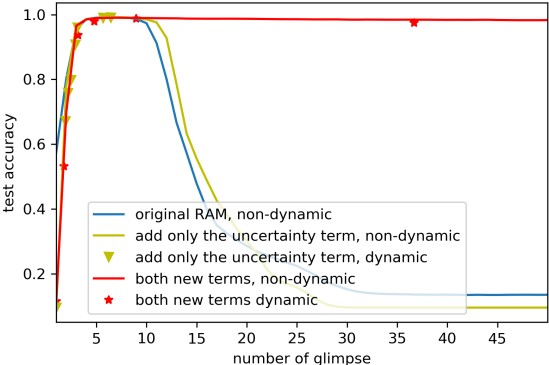

Figure 2: Performance when given diffrent number of glimpses

One may argue that the given longer training time or other hyperparameter tuning, RAM will eventually reach a point where it can give stable prediction accuracy on more glimpses, and the new objective only make it converge faster to that point. But during our experiments, we find with $\lambda_2 = 0.1$ the $J_{intrinsic}$ term can effectively stablize the prediction given more glimpses, even when trained for only 1 epoch. We observe that the l2-norm of internal states of orginal RAM becomes very large given a longer sequence of glimpses while the modified RAM with $J_{intrinsic}$ remains stable.

**Conclusion.**    We revisted RAM from an active sampling perspective (Gottlieb, 2018). The solution to three weaknesses of the orignal RAM is to add two very simple extra terms to the objective function. We believe the attentional learning for actively obtaining new information for decision making has a high potential for both AI and neuroscience research.

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
