# OpenReview forum: "Revisit Recurrent Attention Model from an Active Sampling Perspective"
_NeurIPS.cc/2019/Workshop/Neuro_AI — Real Neurons & Hidden Units @ NeurIPS 2019 Poster_

### Official Review · AnonReviewer3 · 2019-09-23
**Modifications to Recurrent Attention Model inspired by active sampling in neuroscience improves convergence and generalization.**

**Clarity:** 4

**Comment:**

Strengths:

The additional objectives inspired by neuroscience make convergence faster on training accuracy and increase test accuracy on MNIST. They also provide a system to dynamically control the number of glimpses required to make a decision.

Areas for improvement:

I would like to see either a more convincing rationale for your sparsity objective, or an objective that directly addresses the intrinsic utility of an outcome. I appreciate that MNIST may not be the best task for this, and also that utility will be task- and agent-specific.

**Category:**

Neuro->AI

**Clarity Comment:**

The motivation, method, and results of the paper were well written and easy to follow. The only difficulty I had was in reading figure 1, which was excessively small.

**Evaluation:**

3: Good

**Importance:**

3: Important

**Importance Comment:**

Active sampling presents a framework for improving the efficiency of artificial neural networks in tasks requiring interaction between an agent and an environment. Approaches like this are important to try to reduce the training time needed and online processing requirements for artificial agents to make decisions in the real world.

**Intersection:**

4: High

**Intersection Comment:**

This paper is very clearly at the intersection of neuroscience and artificial intelligence, using a well-defined theory from neuroscience to improve a popular model in the AI literature.

**Rigor Comment:**

The authors present their interpretation of Gottlieb's three motives for implementing active sampling. They conclude that the Recurrent Attention Model implements the first of these three (increase expected reward), and propose objective functions that achieve the remaining two: 1) reducing the uncertainty of belief states, and 2) something related to the intrinsic utility or disutility of anticipating a positive or negative outcome.

The quote from Gottlieb (2018) outlining these objectives leaves a lot to interpretation, but the authors present a reasonable method for reducing the uncertainty of belief states that has the bonus feature of providing control over the number of glimpses required to make a decision.  However, I do not see how the belief in the sparsity of the output is related to the utility of making a prediction.

Nevertheless, the authors show that their new objective function improves the convergence of the recurrent attention model Both new terms in isolation improving convergence individually, although the output sparsity objective appears to do most of the heavy lifting. They also show that by using their uncertainty measure to dynamically determine the number of glimpses they increase test accuracy, most of which seem to be accounted for by minimizing the uncertainty in belief rather than output sparsity.

**Technical Rigor:**

2: Marginally convincing

---

### Official Review · AnonReviewer1 · 2019-09-26
**Neuroscience-inspired recurrent attention model**

**Clarity:** 3

**Comment:**

This paper addresses an important problem in artificial intelligence, in the context of existing techniques, and uses neuroscience as inspiration to propose new techniques.

Greater detail in the crucial paragraphs developing the concepts and computations of J-uncertainty and J-intrinsic would be helpful. In particular, why does the new cost term protect the RAM approach from performance degradation with higher numbers of glimpses? This problem is raised and appears to be addressed in the experimental results, but it is unclear why the insights from neuroscience help make this possible.

In Figure 2, the “both new terms, dynamic” plot does not extend into the regime where performance degradation is most extreme; while this may be a result of the technique used to evaluate the dynamic case, it somewhat undercuts the claim that they dynamic case is roughly equivalent in performance.

**Category:**

Neuro->AI

**Clarity Comment:**

The paper is conceptually easy to follow, although there are several minor spelling / grammar / typographical issues.


**Evaluation:**

4: Very good

**Importance:**

3: Important

**Importance Comment:**

The authors use insights from neuroscience to an important problem in artificial intelligence: the problem of active sampling.


**Intersection:**

5: Outstanding

**Intersection Comment:**

The paper directly applies a conceptual approach from neuroscience to improve an existing, widely cited technique in active image sampling.


**Rigor Comment:**

The paper is transparent and benchmarks its approach against existing approaches.


**Technical Rigor:**

3: Convincing

---

### Official Review · AnonReviewer2 · 2019-09-27
**Exploration of an interesting extension idea is hindered by issues replicating the original RAM results**

**Clarity:** 3

**Category:**

Neuro->AI

**Clarity Comment:**

At a high level, the writing is clear, but some of the technical parts could use a revision pass. e.g. the use of the word "bound" on line 112 is potentially confusion, as is the sentence about merging objectives in line 105. The loss names are also a bit unintuitive.

**Evaluation:**

2: Poor

**Importance:**

2: Marginally important

**Importance Comment:**

Training RAM networks faster and making their execution more flexible is potentially useful, but unfortunately I see the experiments provided as too weak to support this paper's approach.

**Intersection:**

3: Medium

**Intersection Comment:**

This is relevant to the workshop, but it's more of a psychology-inspired approach to a solving machine vision problems than a bridge between ML and neuroscience, and it would fit in about as well e.g. in the main conference at CVPR as it would here.

**Rigor Comment:**

Unless I'm missing something, it looks like the authors fail to replicate the setup they are trying to extend. Fig. 1 shows a baseline training error rate of 20%. The original RAM paper reports a test error rate of 1%, and linear regression yield an error rate of around 9%, so to me this points to a bug. Confusingly, Fig. 2 shows a baseline training error rate of 5%. Since these are both inconsistent and so far from the original performance measurements, it makes interpreting the extensions' performance measurements very difficult. Also, since the authors refer often to the original paper but never mention this very large performance disparity, the omission seems borderline dishonest.

**Technical Rigor:**

1: Not convincing

---

### Author Response · Authors · 2019-10-03
**Response to the reviewer**

(updated Oct 3)

We thank reviewers for their comments. We here talk about some concerns from the reviewers and what we want to revise towards the final version.
In brief:
1) To reviewer 1&3, we will elucidate more about the conceptual idea of the new objective in section 3 and we will also give an analysis for the emprical effect of the introduced new objective in section 4, given the extra one-page limit for the final version.
2) To reviewer 2, we will clarify the concerns of replication issues.

1) To reviewer 1&3:
Given the extra page, we will give a clearer elucidation, especially for J-intrinsic. Beside that, we will also give an analysis of the effect of two terms and why the modified RAM can do better in the longer or shorter sequence case. Emprically we find that RAM trained with the J-intrinsic term tends to give a more cautious prediciton: during the process of taking glimpses, the prediction of RAM with J-intrinsic after the first glimpse is very random, but increases in a smoother manner as taking more glimpses, while the original RAM is very unstable in longer sequences. We will address this issue in the revision.

2) To reviewer 2:
We are sorry that some implementation set-up are missing but we do not mean to be dishonest.

We run the experiments with only modified objective changed and other factors all controlled. We run for each version with the same hyper-parameter, for the number of epochs. Since the original RAM converges slower, given the same epoch number for training, the original RAM model would have an inferior performance. The number of epochs trained is decided as the number of epoch the modified RAM achieves a decent performance (paper accuracy). We intend to choose this number of epochs for all four options to show the speed of convergence, but now we realize this is not a rigorous decision. If we wish to compare, we should let the original RAM achieve a decent accuracy, not the modified version. So we rerun the experiments with the number of epochs when the original RAM achieves a good performance. The final test error is 0.79% , (accuracy 99.21%), which should even be better than the paper error 1.29% in the original RAM paper (https://arxiv.org/abs/1406.6247) on MINST when N=6. However, we want to emphasize that for figure 2,  the focus is not the absolute performance, but rather the generalization of more or less glimpses than the training glimpse number.

We will update the experimental results, and also run it for another harder dataset for the revision.

---

### Decision · Program_Chairs · 2019-10-02

Accept (Poster)